# Plant-Based Dairy Alternatives—A Future Direction to the Milky Way

**DOI:** 10.3390/foods12091883

**Published:** 2023-05-03

**Authors:** Diana Plamada, Bernadette-Emőke Teleky, Silvia Amalia Nemes, Laura Mitrea, Katalin Szabo, Lavinia-Florina Călinoiu, Mihaela Stefana Pascuta, Rodica-Anita Varvara, Călina Ciont, Gheorghe Adrian Martău, Elemer Simon, Gabriel Barta, Francisc Vasile Dulf, Dan Cristian Vodnar, Maria Nitescu

**Affiliations:** 1Life Science Institute, University of Agricultural Sciences and Veterinary Medicine Cluj-Napoca, 400372 Cluj-Napoca, Romaniadan.vodnar@usamvcluj.ro (D.C.V.); 2Faculty of Food Science and Technology, University of Agricultural Sciences and Veterinary Medicine Cluj-Napoca, Calea Mănăștur 3-5, 400372 Cluj-Napoca, Romania; 3Faculty of Agriculture, University of Agricultural Sciences and Veterinary Medicine Cluj-Napoca, Calea Mănăștur 3-5, 400372 Cluj-Napoca, Romania; 4Department of Preclinical–Complementary Sciences, University of Medicine and Pharmacy “Carol Davila”, 050474 Bucharest, Romania; 5National Institute for Infectious Diseases “Prof. Dr. Matei Bals”, 021105 Bucharest, Romania; mnitescudsp@gmail.com

**Keywords:** dairy, fermentation, food processing, legume, milk, nutritional value, nuts, sustainability, vegetarian

## Abstract

One significant food group that is part of our daily diet is the dairy group, and both research and industry are actively involved to meet the increasing requirement for plant-based dairy alternatives (PBDAs). The production tendency of PBDAs is growing with a predictable rate of over 18.5% in 2023 from 7.4% at the moment. A multitude of sources can be used for development such as cereals, pseudocereals, legumes, nuts, and seeds to obtain food products such as vegetal milk, cheese, cream, yogurt, butter, and different sweets, such as ice cream, which have nearly similar nutritional profiles to those of animal-origin products. Increased interest in PBDAs is manifested in groups with special dietary needs (e.g., lactose intolerant individuals, pregnant women, newborns, and the elderly) or with pathologies such as metabolic syndromes, dermatological diseases, and arthritis. In spite of the vast range of production perspectives, certain industrial challenges arise during development, such as processing and preservation technologies. This paper aims at providing an overview of the currently available PBDAs based on recent studies selected from the electronic databases PubMed, Web of Science Core Collection, and Scopus. We found 148 publications regarding PBDAs in correlation with their nutritional and technological aspects, together with the implications in terms of health. Therefore, this review focuses on the relationship between plant-based alternatives for dairy products and the human diet, from the raw material to the final products, including the industrial processes and health-related concerns.

## 1. Introduction

Sales of plant-based dairy alternatives (PBDAs) have grown in the last decade and are predicted to increase. PBDAs currently account for 7.4% of the overall milk market share; by 2023, that percentage is anticipated to touch over 18.5% [1]. Sales of plant-based milk alternatives (PBMAs) rose by 9% in 2018, while those of cow milk fell by 6%. Sales of plant-based coffee creamer, which increased by 131%, were the main factor in the growth of other PBDAs [2,3].

The marketing of PBDAs frequently emphasizes sustainability, a dedication to environmental protection, the abolition of “unnatural foods”, or the ethical treatment of animals, which may give them an advantage over conventional dairy products [3]. On-package, in-store, and digital media promotion for PBDAs emphasize is on how these new products are distinct enough from dairy to meet consumer concerns while similar enough to deliver the experience people anticipate and serve as a straight replacement [4]. Dairy goods are starting to market with related claims as worries about sustainability increase; for instance, 21% of dairy products in 2018 launched with the term “grass-fed” [1]. An August 2018 Mintel poll indicated that 27% of internet users 18 and older were willing to pay more for cheese made with milk from “free-range” cows, while 49% were concerned about the environmental impact of dairy farming [5]. Consumers who buy dairy and nondairy products participated in a conjoint survey, including a means-end-chain interview conducted by McCarthy et al. in 2017. They discovered that the plant-based attribute caused moral reactions based on animal treatment and environmental repercussions, a value ladder among nondairy consumers [6]. On the other hand, Pua et al. (2022) highlight the need for a larger variety of PBDAs that have to be studied, offering organoleptic and nutritional improved products [7].

According to Pelletier et al. (2013), young individuals who valued sustainable food production more frequently had better-eating habits overall [8]. This suggests that the ideas of sustainability and a healthy diet may be related. According to Verain et al. (2016), Dutch consumers across various food categories, including the general dairy category, perceive sustainability and healthiness as working in concert [9]. This finding raises the possibility of the halo effect, in which perceptions of sustainability and healthiness are positively correlated. Additionally, it has been discovered that consumer ideals of sustainability, health, and naturalness overlap when it comes to organic products [10]. Even though there is little evidence to support this claim, consumers tend to believe that organic dairy products are naturally healthier, more natural, better for the environment, and better for animal welfare than their conventional counterparts [11,12,13].

As can be seen in Figure 1, a summary of the keywords’ appearance and their connection according to the occurrence was obtained using the Web of Science database (www.webofscience.com (accessed on 9 January 2023)), applying the last 10 years as a filter. A list of 53 keywords was selected, such as property, dairy, vegetarian, sustainability, lactose intolerance (LI), and other correlated words. The figure was obtained using the VOSviewer program in order to obtain an overview of PBDAs and the main aspects studied in this regard. This figure highlights the fact that most of the studies carried out on PBDAs focus on technological production processes, physical-chemical qualities, and consumer acceptance, but also health implications, as described above.

Nutrient sources in food have not been extensively discussed in studies concentrating on vegetarian and vegan diets [14,15]. For instance, vegetarians and vegans may develop ways to replace foods high in animal protein with those high in plant protein, thereby satisfying their protein needs. Increased intake of meat substitute foods, such as tofu and processed textured soy protein foods, would affect these measures [15]. Even though these items are becoming more prevalent in the food market [16], there is currently no information on how much meat alternatives vegetarians or vegans consume [17].

All stages of life can benefit from a well-planned vegan or vegetarian diet. However, vegetarian diets during pregnancy and lactation, as well as infancy and youth, require specific attention. Those who eat a vegan diet should especially be aware of this. There are specific regulations with specific rules in every country regarding pregnancy, newborn, and child growth, such as interdiction or followed under medical supervision [18,19,20]. Referring only to the vegetarian or vegan diet as a lifestyle, the purchase, and consumption of PBDAs is an easier activity for people who follow this diet because nutritional education is directed only towards this sector and they pay more attention to deficiencies that can appear [21,22].

A large number of existing studies in the broader literature have examined the PBDAs’ effects on human health, but more research regarding in vivo studies needs to be done. First of all, this study highlights the starting point in the manufacture of PBDAs, presenting the nutritional profile and the effects of the nutrients contained, going further with the obtained products and the technology involved in the processing. At the end of this manuscript, we present the implications of these alternatives in human health from a neutral point of view to complete the picture of the correlation between food and human health.

From a legislative point of view, according to Regulation (EU) No. 1308/2013 of the European Parliament and the Council of the European Union, the term “milk” can be used for “milk treated without altering its composition or for milk the fat content of which is standardized under Part IV” (Part IV—Milk for human consumption falling within CN code 0401) or “in association with a word or words to designate the type, grade, origin and/or intended use of such milk or to describe the physical treatment or the modification in composition to which it has been subjected, provided that the modification is restricted to an addition and/or withdrawal of natural milk constituents”. An important aspect that should be mentioned regarding milk is the animal species, as stated in Part III Milk and milk products, from Regulation No. 1308/2013, as follows “as regards milk, the animal species from which the milk originates shall be stated, if it is not bovine” [23].

The current review’s objective was to look into nutritional factors associated with health benefits related to PBDA consumption, including the prevalence of dietary nutrient needs. This study also examined the technological point of view, from raw materials to the end products, including the nutritional profile.

### Experimental Method—Literature Research

The manuscript was done following the Preferred Reporting Items for Systematic Reviews and Meta-Analyses (PRISMA) statement. All authors independently searched the literature, and any discrepancies were settled through consensus [24].

We analyzed for potentially relevant publications in the scientific electronic databases Scopus, Web of Science Core Collection (WoS CC), and PubMed. We created a methodical search approach that included the term “plant-based dairy alternatives”. For all three databases, we used the same search method. The inclusion criteria for this review were studies published in the last 10 years (2013–2023), English language, and human and rodent studies. Studies that include correlated aspects between PBDAs and the nutritional profile of raw materials and final products, technology, and consumers’ acceptability, but also correlations with health-related aspects, were included. The exclusion criteria used in the paper were related to articles that included animals other than rodents, and studies that did not provide data about PBDAs or nutritional, technological, and health aspects. All 354 abstracts were screened separately by all authors. We narrowed the pool of possibly relevant articles to 194 by applying inclusion/exclusion criteria to the data in the abstract (Figure 2). Using the same inclusion and exclusion criteria as for the selection of the abstracts, we examined the retrieved full-text articles. The writers discussed discrepancies during the selection process until a unit consensus was formed. After excluding 46 articles that did not meet the inclusion criteria, we ultimately decided to include 148 full-text articles in our review. In Figure 2, a flow chart illustrating the specific steps of the systematic review article selection process is provided.

## 2. Dietary Sources of PBDAs—Description and Nutritional Profile

Consumer demand for plant-based products, such as plant-based beverages as an alternative to milk, is constantly growing. These dairy alternatives are made from a variety of plant sources, such as cereals (oat, rice, corn, spelt), pseudocereals (quinoa, teff, amaranth), legumes (soy, peanut, lupine, pea, and chickpea), nuts (almond, coconut, hazelnut, pistachio, walnut), and seeds (sesame, flax, hemp, sunflower) [25].

### 2.1. Cereals and Pseudocereal Sources

Whole-grain cereals are regarded as an essential component of the daily diet, having great health advantages. Oats are among the most nutritious cereals due to their high levels of micronutrients and macronutrients (Table 1) [26,27]. Oat cereal belongs to the *Gramineae* family. The most widely farmed types of oats are hulled oats (*Avena sativa* L.) and naked oats (*Avena nuda* L.). The mature oat grains are composed of four structural components: husks, bran, endosperm, and germ [28]. The oat husk is mechanically removed from the oat groat during the industrialization process for further human consumption. The outer layer of the oat grain accounts for 30% of the dry mass and contains the highest concentration of micronutrients such as B vitamins, minerals, and phytochemicals [28]. The endosperm accounts for 55–70% of the groat’s weight and is mainly composed of starch (70–78%), protein (9–12%), fat (6–8%), and dietary fibers (4–6%) [28].

Rice (*Oryza sativa* L.) is a cereal in the *Poaceae* family and is presented in monocotyledonous form. Rice cultivation is widespread, with over 480 million metric tons produced each year globally. Rice has a starchy structure that accounts for 78% of its composition, followed by protein content, which accounts for 6–7%, and reduced amounts of lipids, vitamins, and minerals [29,30].

Sweet corn is also an ingredient found in plant-based dairy alternatives. The well-known nutritional drink made from sweet corn is the most widely produced product from the corn kernel. Producing sweet corn beverages includes operations such as grinding, filtering, mixing, homogenization, filling, and sterilization [31].

Furthermore, pseudocereals are recognized as excellent sources of ingredients for the development of PBDAs. They have attracted attention due to the high concentrations of valuable components to human health, such as proteins, peptides, flavonoids, phenolic acids, fatty acids, vitamins, amino acids, dietary fiber, lignans, and unsaturated fatty acids [32]. Pseudocereals (quinoa, amaranth, and buckwheat) are often found among the ingredients used in the preparation of various foods, such as soups, tortillas, alcoholic beverages, and plant-based dairy alternatives. Pseudocereals are most commonly used in the form of flour, and their incorporation into food products needs a series of analyses regarding aspects such as the technological and organoleptic features of the resulting food products [32]. Because of their high protein content, outstanding nutritional value, and low cost, quinoa proteins have been identified as feasible alternatives for producing plant-based dairy products [33].

Quinoa (*Chenopodium quinoa Willd*) contains a high gluten-free protein content, as well as other valuable nutrients such as vitamins, minerals, dietary fiber, and unsaturated fatty acids [34]. Quinoa protein has a higher protein concentration than cereal protein, and it has the optimal amino acid balance, characterized by its high lysine level [35]. An important aspect that must be mentioned is related to the content of inhibitory compounds with a negative effect on the absorption of nutrients [34]. Among the inhibitory compounds are polyphenols, tannins, and phytates, which decrease the concentration of absorbed essential minerals. Phytates operate as mineral chelators (zinc, iron, and calcium) and produce a macromolecule that the organism cannot digest or absorb in the human digestive tract [34]. The amount of phytate in raw quinoa ranges from 7.92 to 8.93 g kg^−1^; therefore, processing techniques such as roasting, boiling, fermenting, and soaking are required to minimize antinutritional effects in plant-based foods [34].

Amaranth is another pseudocereal included in the ingredient list of PBDAs. Amaranth seeds are notable for their high protein content but also include considerable amounts of lipids, carbohydrates, and dietary fiber (Table 1) [32,36]. The amino acid composition of the proteins identified in amaranth seeds is similar to that recommended composition by the World Health Organization for a high biological and nutritional value [32].

Buckwheat is a pseudocereal that is also utilized in the plant-based dairy sector. Before consumption, buckwheat seeds require a dehulling operation, because the seed shell is indigestible. Roasting is the primary treatment used to produce buckwheat products, and it has been found to enhance the concentration of various phytosterols in grains and groats, such as campesterol, sitosterol, avenasterol, D-7 stigmasterol, and cycloartenol, while decreasing the content of stigmasterol [32]. Moreover, valuable concentrations of phenolic compounds (275.5–532 mg gallic acid equivalent (GAE)/100 g dw) were reported by G. Rocchetti *et al.* [37].

Therefore, due to their complex nutritional profile, cereals and pseudocereals are well-recognized and appreciated grains used as ingredients in producing PBDAs (Table 1 and Table 2).

### 2.2. Legume-Based, Nut-Based, and Seed-Based Sources

These foodstuffs are rich in proteins, have high nutritive value, and present a healthy alternative capable of decreasing the possibility of stroke and several heart-related diseases [38,39]. The most noteworthy examples of legume sources comprise mainly soy, peanuts, lupins, peas, chickpeas, lentils, and alfalfa [25].

Within these sources, soy is the most prevalent legume, soy drink being the main alternative extracted from soybeans (*Glycine max* L.), *a* member of the *Fabaceae* family with global production in 2020 of above 353 million metric tons [40,41,42]. As can be observed in Table 1, soy has high protein, vitamins, fatty acid (mono- and polyunsaturated), oil, and amino acid content, and strong antioxidant activity due to the phytoestrogens and isoflavones (genistein, daidzein, and glycitein) present in its structure [43,44,45]. To improve the digestibility and adsorption of soy products and also the sensorial aspects of soy-based foods, one of the best methods is fermentation. Through this microbial decomposition, the oligosaccharide (stachyose, and raffinose), and antinutrient (urease, trypsin inhibitor, and phytic acid) contents are decreased, contributing to the organoleptic properties and improving the bioavailability of bioactive compounds [46,47].

Another controversial legume source belongs to peanuts, otherwise known as groundnut (*Arachis hypogaea* L.), a member of the *Leguminosae* family that has a low cost and is strongly nutritious but is also a prevalent allergen (with 12 recognized allergens) [48,49]. Besides being consumed in childhood or adulthood as a snack, peanuts are an important source of oil and are frequently used as pastes, butter, or other milk alternatives [50,51]. Encompassing around 70% of high-quality proteins, categorized as globulins (saline soluble—arachin and conarachin I and II), albumin (water soluble), and also with an important bioactive compound, namely resveratrol, this species has a positive effect in inflammation, cardiovascular disease, type 2 diabetes, and cancer reduction [52]. Although peanuts are the most caloric oilseeds (567 kcal), they can still be applied in weight control, as they confer sustained satiety given the enhanced oil (27–29%), fiber, and protein content, and after roasting they are not bioaccessible in the course of digestion [48]. Roasting (dry roasting, oil roasting) peanuts confers valuable effects besides flavor and prolonged shelf life and generates volatile and aroma-active compounds [48,53].

Another valuable gluten-free legume source belonging to the *Fabaceae* family is the *Lupinus* species [54]. This crop has a continually diminishing total global production reaching 1.05 million metric tons in 2020 even though it is easily cultivated on unproductive soils [55], is inexpensive, is a high-protein food, and has increased fiber content with a reduced glycemic index [56,57]. Even though it is protein-rich, this legume is highly bioavailable, as evidenced by Mariotti et al. (2001) [58]. With increased phytochemical content, such as alkaloids, phytosterols, tocopherols, polyphenols, and bioactive peptides, lupin presents good perspectives in functional food production [59].

Peas (*Pisum sativum*) and chickpeas (*Cicer arietinum* L.) are also green alternatives for vegetable-originated food products with a global production of 14.64 and 15.08 million metric tons, respectively [60,61]. They have cheap production and enhanced protein content, such as albumin, globulin, tryptophan, and lysine [62,63]. Chickpea is also rich in enzymatic functional oligopeptide fragments and bioactive compounds such as isoflavones (genistein, ononin, trifolirhizin, sissotrin, calycosin, biochanin A, and formononetin), which provide estrogenic, antioxidant, antimicrobial, and antifungal features, and afterward can be considered as a medicinal ingredient [64,65]. 

Almonds (*Prunus amygdalus, Prunus dulcis*) are the first in tree nut production globally [66]. This nut is mainly grown in the Mediterranean region as a snack and for functional food production because they are high in nutrients (macro- and micronutrients), and rich in lipids and proteins [67]. The average daily consumption of nut-based sources provides important dietary fiber, phytosterol, and vitamin E content, with beneficial health-related effects (cardio-protective, metabolic, cancer, blood cholesterol reduction, and other beneficial effects) [68]. The almond fruit also contains important phenolic compounds (caffeic, p-coumaric, vanillic, and ferulic acids) and flavonoids (procyanidins, kaempferol, quercetin, delphinidin, etc.) [68].

Hazelnuts (*Corylus avellana* L.) are also rich in vitamins (soluble—B, C, E, calcium, phosphorus, potassium, and magnesium; insoluble α-tocopherol, retinol, δ-tocopherol, etc.), fatty acids (oleic, linoleic, linolenic, palmitic, stearic, etc.), aminoacids (essential—arginine, leucine, histidine, izoleucine, lysine, methionine, threonine, valine, phenylalanine; nonessential—glutamine, asparagine, alanine, glycine, tyrosine, serine, proline) [69]. Being nutritious, with important antioxidant effects, rich in fibers, and high in vitamins, it is recommended for consumption based on several health effects [70]. 

Walnuts (*Juglans regia* L.), as with almonds and hazelnuts, belong in the category of “brain-food”, as they bequeath mental alertness, memory, and concentration, and upgrade sleep quality [71]. These qualities are attributed to the high content of fatty acids (MUFA, PUFA, especially n-3 PUFAs), vitamins (B, E), essential minerals (calcium, copper, magnesium, manganese), phenolic acids, flavonoids, dietary fiber, and other components that are health supportive [72,73]. These nuts (almond, hazelnut, and walnut) provide great satiety, but their digestibility is low and only approximately 20–21% of their energy is bioaccessible, the production of dairy products from them could increase their bioaccessibility and bioavailability [74].

Part of the *Anacardiaceae* family, pistachio (*Pistacia vera* L.), is also rich in fatty acids, proteins, and dietary fiber, it is nutrient dense and has relevant healthy components, and from these, it has the highest anti-inflammatory and antioxidant activity [75]. The protein digestibility corrected amino acids score (PDCAAS) of roasted pistachio is higher than the PDCAAS of raw pistachio (81 and 73), while the digestible indispensable amino acids score (DIAAS) of roasted pistachio is smaller than the DIAAS of raw pistachio (83 and 86); thus, pistachio nuts can be considered a good protein source [76]. In a study by Baer et al. (2011) [77], the gross energy of this nut ranged between 29 and 46 kJ/g, which is higher than the recommended energy density of a healthy diet; however, it also indicated that these nuts do not contribute to a higher BMI (body mass index) value, reduce the overall BMI (lipids absorbed inadequately), and provide additional beneficial effects on health [78].

Coconut (*Cocos nucifera* L.), part of the *Aracaceae* family, is composed of husk, copra, and water, each with important component concentrations. Copra is rich in potassium and phytochemicals, and from pressing copra coconut milk is obtained, which is a nutrient-dense liquid [79,80]. As presented, nut-based sources provide important nutrients, phytochemicals, and bioactive compounds that are essential in the human diet and provide multiple positive health effects [25,81].

The last section analyzed the PBDAs belonging to seed-based sources, such as sesame, flax, hemp, and sunflowers. Sesame has a smooth flavor and a considerably aromatic odor, abundant in protein, fat, vitamins, minerals, and dietary fibers; thus, it presents a superior nutritional advantage [82]. The recently discovered sesaminol (lignan), has increased antioxidant features, while sesamin and sesamolin have almost no antioxidative effect, but they exert proantioxidant activity (powerful antioxidants through in vivo metabolic alterations) [83,84]. Sesame has efficient digestibility, and through the consumption of 1–3 g of sesame daily, important health-promoting effects can be obtained [85].

Flaxseed (*Linum usitatissimum*) is plentiful in fibers, lignans, antioxidants, superior quality proteins, and α-linoleic acid, and can exert various therapeutic effects [86]. Moreover, has high contents of oil, fiber lignin, omega-3/omega-6 fatty acids, and other valuable compounds [87,88]. Meanwhile hempseed (*Cannabis sativa* L.), part of the *Cannabaceae* family is abundant in oil, fiber, proteins, carbohydrates, vitamins (B1, B2, B3, B6, C, D3, E, carotenoids, magnesium, and micro- and macro-elements), and phytochemicals such as cannabinoids with the two most representative ones being (–)-trans-Δ^9^- tetrahydrocannabinol (THC), with intoxicating effects (this is found in low concentration (<0.3%), in industrial hemp), and (-)-cannabidiol (CBD) [89]. These kinds of nuts sustain the immune system, protect against oxidative stress, help in cardiovascular diseases and diabetes, reduce inflammation, and have anticancer effects [90,91]. 

Sunflower (*Helianthus annuus* L.) is also rich is omega-3 and omega-6 fatty acids, linoleic and oleic acids, phytosterols, vitamins (B, E, folate, and niacin), minerals, flavonoids, and antioxidants [92]. Its oil and seeds are part of the *Astraceae* family and are frequently used in frying, cooking, and baking [92,93]; additionally, the by-product of sunflower also contains several nutrient-rich substrates such as cake and meal [94,95].

**Table 1 foods-12-01883-t001:** Nutritional profile of dietary sources for plant-based dairy products.

Dietary Sources (g/100 g)	Water	Energy	Protein	Total Lipid	Ash	Carbohydrate	Total Dietary Fiber	Total Saturated Fatty Acids	Total Monosaturated Fatty Acids	Total Polyunsaturated Fatty Acids
Oat [27]	8.22	389 kcal	16.90	6.90	1.72	66.30	10.60	1.22	2.18	2.54
Rice [30]	12.90	360 kcal	6.61	0.58	0.58	79.30	-	0.16	0.18	0.15
Corn [96]	10.40	365 kcal	9.42	4.74	1.20	74.30	7.30	0.67	1.25	2.16
Quinoa [97]	13.30	368 kcal	14.10	6.07	2.38	64.20	7.00	0.71	1.61	3.29
Teff [98]	8.82	367 kcal	13.30	2.38	2.37	73.10	8.00	0.45	0.59	1.07
Amaranth [36]	11.30	371 kcal	13.60	7.02	2.88	65.20	6.70	1.46	1.68	2.78
Buckwheat [99]	9.75	343 kcal	13.20	3.40	2.10	71.50	10.00	0.74	1.04	1.04
Soy [47]	7.25	327 kcal	51.46	1.22	1.30	33.92	17.50	0.14	0.21	0.53
Peanut [100]	6.50	567 kcal	25.80	49.20	2.30	16.10	8.50	6.30	24.40	15.60
Lupin [57]	5.52	371 kcal	92.60	-	5.30	11.00	<0.10	0.48	0.25	0.20
Pea [101]	78.90	81 kcal	5.42	0.40	0.87	14.40	5.70	0.07	0.03	0.19
Chickpea [64]	60.2	164 kcal	6.04	2.59	0.92	62.95	7.60	0.60	1.38	2.73
Almond [67]	4.70	575 kcal	21.22	49.42	2.99	21.67	12.20	3.73	30.89	12.07
Hazelnut [70]	5.31	628 kcal	15.00	60.70	2.29	16.70	9.70	4.46	45.70	7.92
Walnut [102]	3.62	630 kcal	16.66	66.90	1.81	13.70	6.70	6.13	8.93	47.20
Pistachio [75]	4.37	560 kcal	26.16	45.40	2.99	29.00	10.60	5.60	24.50	13.30
Coconut [79]	47	354 kcal	3.33	33.50	0.97	15.20	9.00	29.70	1.42	0.37
Sesame [82]	4.69	573 kcal	17.60	49.70	4.48	9.85	14.90	7.09	18.80	21.80
Flaxseed [86]	-	536 kcal	20.00	42.90	-	29.00	28.00	3.57	7.02	24.30
Hempseed [103]	4.96	553 kcal	31.60	48.80	6.06	8.67	4.00	4.60	5.40	38.10
Sunflower [93]	4.73	584 kcal	20.80	51.50	3.02	20.00	8.60	4.46	18.50	23.10

**Table 2 foods-12-01883-t002:** Beneficial and antinutritional effects of PBDA sources’ compounds.

Dietary Sources		Compound	Concentration	Beneficial and Antinutritional Effects	References
Cereals	Oat (mg/100 g)	Phytate	278.7	Good source of valuable nutrients that can considerably contribute to human diet and nutrition; Antinutritional factors limit overall nutrient absorption, particularly minerals, proteins, and vitamins	[104]
		Tannin	44.7		
		Oxalate	48.4		
	Rice (g/kg)	Phytic acid	21.03	High phytic acid consumption has been associated with deficits in Zn and Fe	[105]
	Corn (mg/100 g)	Polyphenols	425.8	Antinutritional compounds such as phytic acid, polyphenols, and tannins can limit protein and carbohydrate bioavailability and digestibility by forming complexes with minerals and by inhibiting enzymes	[106]
		Tannins	215.1		
		Phytates	278.7 g		
Pseudocereals	Quinoa	Saponins	1.63 mg/g	Bitter taste; Reduced mineral bioavailability	[107]
		Phytic acid	375.27 mg/100 g		
		Tannins	3.41 mg TaE/100 g		
	Buckwheat (g/100 g)	Phytic acid	18.07	Affect small intestine metabolism, disrupt starch and protein digestion, and reduce mineral absorption	[108]
		Trypsin inhibitor	5.94		
		Tannin	5.13		
		Saponin	3.23		
	Amaranth (mg/100 g)	Tannin	1.50–3.46	Reduced nutrients bioavailability	[109]
		Oxalate	3.73–6.81		
		Saponin	2.94–4.962		
Legumes	Mung bean (mg/100 g)	Phenols	238	Developing functional diets; Use in the treatment of diseases including cancer and cardiovascular disease	[110]
	Soybean	Trypsin	1952 U	Biochemical usefulness	[111]
		Agglutinin	6400 HU		
	Lupin (mg/g)	Phytic acid	0.29–0.52	Improved in vitro protein digestibility;	[112,113]
		Saponins	0.85–2.75		
		Tannins	0.61–1.34		
		Amylase inhibitor	85.63–182.9		
	Pea (mg/100 g)	Saponins	2.97	Bitter or unacceptable taste; Causes flatulence; Decreased protein digestibility	[114]
		Phytic acid	5.76		
		Oxalate	3.44		
		Alkaloids	6.97		
		Cyanide	2.3		
Nuts	Hazelnut (mg/g)	Phenolics	8.71–12.9	Have a negative impact on feed intake, body weight increase, and feed conversion	[115]
		Phytate	18.5–33		
	Almond (mg/100 g)	Hydrogen cyanide	21.6	The absorption of minerals, such as calcium, iron, zinc, and magnesium can be affected	[116]
		Oxalate	26.4		
		Tannin	39.4		
Seeds	Sesame (mg/100 g)	Tannin	5.62	Enzyme binding; Binding of feed components such as proteins or minerals; Digestion processes limitations	[117]
		Phytin	25.05		
		Saponin	4.97		
		Oxalate	15.66		
	Hempseed (g/100 g)	Phytate	4	Formation of insoluble calcium oxalates; Reduced mineral element bioavailability	[118]

U: units; HU: Hounsfield units.

As highlighted previously, these plant-based (cereals, pseudocereals, legumes, nut-based, and seed-based) dietary sources for the production of dairy products are rich in nutrients, proteins, and minerals, and have a low-glycemic index. Although there is a need for further studies, especially with respect to their shelf life, stability, and challenges in processing, they are still a more healthy alternative and have a lower environmental impact than the usual dairy products [25,39,119,120].

## 3. Plant-Based Dairy Alternatives

The demand and supply of products that substitute dairy products are continuously increasing. There is a varied range of vegetable products with an improved nutritional profile, from milk substitutes to desserts containing dairy products. All these products can also be found with vegetable alternatives to reach several population groups, from vegetarians to different pathologies that require these options.

### 3.1. Plant-Based Milk Alternatives (PBMAs)

Consumption of bovine milk, especially cow milk, plays a crucial role in the nutrition of humans of all ages since the 7th millennium BC [121]. Milk is a complex food with high nutritional value that is important for human health. Milk provides high biological value proteins, lipids, vitamins, and minerals, especially calcium. However, not all human bodies react well to milk consumption. LI (affecting about 75% of the world’s population) and cow’s milk protein allergy are the main disadvantages of milk use. Furthermore, the vegan diet is associated with a healthy lifestyle throughout modern society and animal-origin food is eliminated for several reasons (e.g., to reduce environmental pollution) [122]. Therefore, PBMAs have gained the considerable potential to replace mammalian milk in daily diets. PBMAs, also known as “vegetable milk” or plant milk, use cereals, pseudocereals, legumes, nuts, and seeds as raw materials. PBMAs are water-soluble extracts that resemble bovine milk in appearance. They are manufactured by reducing the size of the raw material, followed by extraction in water with homogenization, separation of the solid phase from the liquid phase, and formulation of the final product [122].

Production of PBMAs follows some common and specific steps, depending on the processed raw material. The common steps in plant milk production are wet milling, filtration, the addition of supplementary ingredients, sterilization, homogenization, aseptic packaging, and cold storage. As supplementary ingredients, we can mention gums (for improved plant milk stability), salt, sugar, oils (for improved sensory properties), minerals, and vitamins [81,122]. Specific steps for PBMA manufacturing are dehulling, roasting, dry grinding, steeping in dilute acid, the addition of some enzymes, and soaking in deionized water. An overview of each above-mentioned technological step used in the production of 12 different PBMAs was recently revised by Aydar et al. [81]. Fresh PBMAs have a very short shelf life and their consumption is limited. The most common preservation method for both bovine milk and PBMA production is a thermal treatment, especially pasteurization and ultra-high temperature (UHT). However, thermal treatment can cause changes in the properties of proteins, vitamins, and minerals, thus influencing the stability and sensory properties of the final product. Regardless, innovative nonthermal techniques are being researched to increase the shelf life of PBMAs without affecting their stability, nutritional, and sensory profile [122]. Ultrasound, pulsed electric fields, ohmic heating, and high and ultra-high-pressure homogenization are a few examples of sustainable techniques for PBMA preservation that could be improved for large-scale production. However, there is a gap of knowledge between the research field and industrial scaling up since experiments of such novel technologies have been performed on only a few plant sources, generally almond and soy milk substitutes [81]. Another challenge for PBMAs is represented by a higher cost of production. Cost is a very important factor in food choice decisions. Investigating the non-sensory characteristics of the PBMAs in a Canadian region, many of the participants felt these products are expensive [123]. Regarding the environment, the manufacturing of PBMAs has positive effects including decreasing the water foodprint and creating the potential for reducing climate change and ecotoxicity [81]. PBMAs can be consumed as a drink or can be processed in products such as cheese, cream, yogurt, butter, ice cream, and other types of sweets.

The nutritional profile of PBMAs is an important aspect of being considered as an alternative to dairy products. PBMAs have a different nutritional profile compared to bovine milk. Furthermore, it is directly influenced by the plant source, processing, and fortification with supplementary ingredients [122]. Plant extracts contain lower nutritional value than raw plants. Generally, PBMAs are nutritionally inferior to cow milk, with specific characteristics. The energy content is less consistent in PBMAs and depends on different brands and raw materials, while different brands of bovine milk with the same fat content have little differences in the provided nutritional energy [121]. Concerning macronutrients, PBMAs have a lower protein content than bovine milk. Additionally, animal proteins present a higher nutritional quality (high variability of amino acids) than plant proteins and greater digestibility. Only soy-based milk presents a protein content comparable to milk protein content. Improved protein content in PBMAs could be obtained by mixing different raw materials. Milk also contains cholesterol and high carbohydrate content. PBMAs are lactose-free and cholesterol-free, having a higher content of unsaturated fatty acids [124]. However, it is important to consider refined sugars present in some PBMAs that posses a higher glycemic index than bovine milk [122]. Furthermore, despite many people having lactose intolerance, natural lactose has been shown to enhance the bioavailability of calcium and other minerals [121]. 

Regarding micronutrients, plant-based milk substitutes have a lower content of minerals and vitamins and their absorption is less consistent than in milk. Generally, plant milk is fortified with vitamins, especially calcium (which is naturally found in milk) to improve its nutritional value. Furthermore, PBMAs contain some antinutrients such as phytic acid, trypsin inhibitors, and inositol phosphates. These compounds interfere with the absorption of minerals and reduce the protein’s digestibility. However, PBMA contains other important components for the human body that are not present in bovine milk, such as isoflavones (especially in soy-based milk alternatives) and dietary fibers. Isoflavones have antioxidant activity and prevent cardiovascular disease, prostate cancer, and osteoporosis [122]. Thus, it is necessary to compare the energy, protein, fat, and calcium contents of PBMAs to bovine milk to examine whether or not PBMAs can effectively substitute for milk in the diet in the provision of micro and macronutrients [121]. Table 3 shows the nutritional profile of some examples of PBMAs and other dairy products are further presented.

Animal protein provides unique sensory and textural properties to foods that are not easily replicated with PBMAs [125]. In the industry, two challenges have been observed concerning sensory acceptability by consumers: a final product having a “beany” or “painty” or a “chalky” mouthfeel caused by insoluble large particles [81]. Consumers identified the different flavors of the PBMAs, as well as the textural attributes and appearance. Additionally, these aspects affect their purchasing decisions and conceptualization. The addition of flavorings and mixing of different raw materials are often used to make PBMAs products more palatable and more sensory appealing. A recently performed survey tried to identify the influence of flavorings on the acceptability of PBMAs. Many participants revealed that they usually choose to drink flavored PBMA products and not unflavored ones. The addition of vanilla and chocolate flavoring had a market effect on the consumers’ liking scores, and the chocolate plant milk was usually more liked than the unsweetened plant milk [123].

During production, different raw materials are mixed in (e.g., rice and oat) to increase the total protein content [81], but these beverages should be studied to better understand their sensory properties [123]. Furthermore, food scientists are currently developing innovative strategies to improve the sensory properties of PBMAs [125]. Nonthermal treatments used to increase the shelf life of PBMAs protect their sensory characteristics and have a more favorable sensory effect on the final product than thermal technologies [81]. For instance, pressure homogenization treatment reduces the degradation risk of nutritional and sensory quality attributes compared to heat treatment [122].

When choosing a PBMA, label claims have a substantial impact. For instance, branded products with vegan-friendly labeling encourage important aspects of society (e.g., ethical concerns about animal welfare) and thus the demand for vegan-friendly labeled products is significantly increased. However, consumers should also pay attention to other aspects. Based on nutritional differences, public education initiatives and labeling requirements have to be performed to inform the public to not take PBMAs as a direct nutritional alternative to bovine milk [121]. Consumers generally focus on the macronutrients of PBMAs because of labeling protocols. Since PBMAs contain micronutrients with valuable bioactivity that are not present in bovine milk, consumers have to be informed about all nutritional consumption and their health effects on the human body [81].

Nevertheless, the amount of these beneficial micronutrients is often not present on the label and is likely to differ between product formulations. Fortified PBMAs with calcium remain a problem due to sedimentation. It was reported that unshaken calcium-fortified soy substitutes amounted to only 31% of their label claim and averaged 59% when shaken. Furthermore, only two of eight fortified soy substitutes met their label claim for calcium when shaken. There is no means of determining what percentage of calcium is resolubilized in beverages after shaking since most PBMAs are packaged in opaque containers [121]. 

Additionally, plant-based milk substitutes are expected to solve the bovine milk allergy and LI, but they may cause other allergies compared to bovine milk. For example, soy proteins can cause allergy and 14% of the people who suffer from cow’s milk allergy also have reactions against soy products [122]. Other common food allergens are nuts, and, therefore, nut-based milk substitutes should be properly labeled. Therefore, labeling PBMAs should carefully inform the consumers of what the packaging contains. Since bovine milk is used as raw material for producing dairy products such as cheese, cream, yogurt, butter, and ice cream, it is expected that plant-based milk substitutes will play the same role for PBDAs. Therefore, some particularities of the main PBDAs (cheese, cream, yogurt, butter, ice cream) are discussed in the next subsections.

### 3.2. Plant-Based Cheese Alternatives (PBCAs)

PBCAs are defined as “edible materials prepared from plant ingredients that are designed to have a similar appearance, texture, and flavor as bovine-based cheeses”. The first PBCA varieties, including fermented tofu, were first produced and consumed in China in approximately the seventeenth century. The production principle of PBCAs is to match the physicochemical and sensory properties of a known conventional cheese (e.g., cheddar). The difference between conventional cheese varieties is in the different processing routes. While producing PBCAs, both ingredient and process selection is required since there are several raw materials. PBCAs are obtained from (i) polysaccharides (especially from starches of different plants), (ii) proteins from legumes (e.g., pea, soy, lupin), potato, nuts, and seeds, zein, and (iii) a mix of solid (e.g., cocoa, coconut, palm oil) and liquid (avocado, canola, sesame, soybean, sunflower oil) fats at room temperature [126]. 

However, soy remains the most commonly used plant for producing PBCAs [125]. Sensory attributes play an important role to increase the acceptability of PBCAs. Recently, Falkeisen et al. [127] performed a survey on consumer perception and emotional responses to PBCAs. Buttery, smooth, and soft attributes increased participants’ liking of PBCAs. Meanwhile, mouth-coating, rubbery, and off-flavor decreased their acceptability. These characteristics were differentiated by the primary raw material. Overall, these products were not well-liked by the panelists (scored very low on the nine-point hedonic scale). However, several strategies have been used to improve the sensory qualities of PBCAs, especially of soy-based cheese, such as modified fermentation methods, blending PBMAs, and modified processing of soybeans. The impact of these methods on consumer preferences and the limitations of applied sensory evaluation methods in research studies were summarized by Short et al. [125]. More research is needed to understand how plant-based ingredients need to be extracted and processed to obtain the desired structure and flavors for increasing the marketability of PBCAs [126].

### 3.3. Plant-Based Cream Alternatives (PBCrAs)

PBCrAs are oil-in-water emulsions that use vegetable fats (typically 30–40%) dispersed in a continuous water phase to imitate dairy creams (milk fat globule dispersion in the aqueous skimmed milk phase). The stability of the emulsions is influenced by different factors such as emulsion droplet size and interactions of ingredients (fat, emulsifiers, and stabilizers) in water and at the interface [128,129]. The possibility to replace milk fat in dairy cream with vegetable fat in PBCrAs by using oleogels is receiving great interest. Oleogels have a solid-like gel structure with more than 90% liquid vegetable oil. Recently, they were successfully used as a fat replacer in producing filling cream [130]. The potential of using bigels as an alternative to cream was also investigated. Cui et al. [131] produced bigels by using glycerol monolaurate, medium-chain triglycerides, and cynnamaldehyde as the oil phase and chitosan as the aqueous phase. After optimization, bigels exhibited textural attributes and appearance that were fairly similar to a commercial cream, with a lower melting point. Bigels are polysaccharide-based edible gels that contain both oil and water as the predominant phase.

Despite this approach offering a huge potential to create PBCrAs, this field is still in the incipient phase of development [132]. Hydrocolloids such as whey protein concentrates, modified starches, and protein–polysaccharides complexes were used as saturated fat replacers in low-fat whipping cream (20–30% fat). The obtained products exhibited comparable physical and textural properties with commercial dairy whipping cream. Hydrocolloids play an important role in maintaining emulsifying capacity, water retention ability, and high viscosity [133]. 

In addition to plant-based fat replacers, plant proteins have also been used as alternatives to animal proteins in PBDAs. Different plant protein isolates (from soy, faba, and pea) at various protein concentrations (2, 3, and 4%) and homogenization pressures were used to make PBCrA emulsions. Results indicated that the homogenization step contributed towards changing the surface-active properties and functionality of the proteins, which leads to a stable cream emulsion [128]. The cream can be used as such or for producing pasteurized cream, whipped cream, cheese, ice cream, and cream liqueur [129]. Thus, the increased demand for vegan products provides an impulse to the manufacturer to find innovative solutions and pave the way for the creation of new products such as PBCrAs.

### 3.4. Plant-Based Yogurt Alternatives (PBYAs)

Due to its important nutritional characteristics, such as calcium, high-quality proteins, PUFA, and an adequate level of isoflavones, which prevent bone degradation and have anticancer effects, the inclusion of PBMAs and its by-products in the diet is raising substantial interest [134,135,136]. Soy, almond, and coconut milk are among the most consumed vegetables in the technology of PBYA manufacturing [135,136]. The preparation of yogurt consists of the lactic acid fermentation of milk by the action of starter bacteria [137]. Unfortunately, PBYAs present texture and stability limitations compared to dairy yogurts. The various textures of commercial PBYAs could be caused by their reduced protein concentrations, and because these proteins do not coagulate as well as casein, gelling agents need to be added [134,135,137]. 

A recent experimental work investigated the impact of inulin (20–70 g/L), as a substrate of thickening agent, on the chemical, physical, and sensory properties of ultrafiltered soy yogurt alternatives. Additionally, the addition of membrane concentration of soy proteins (14.5 ± 0.2 g/100 g) enriches yogurt with more protein (59 g/L), vegetable fat (15 g/L), and less ash and antinutrient content. The finished product had a pleasant aroma, flavor, and color; thus, the yogurt containing 50 g/L of inulin achieved the highest overall acceptability (*p* < 0.05) [136]. Pachekrepapol et al. [134] highlight the use of tapioca starch (0.5, 1.0, 1.5, and 2.0%) as a stabilizer in the production of coconut milk-based yogurt alternatives stored at 4 °C for 14 days. These results indicated that the inclusion of tapioca starch aids in reinforcing the gel network by increasing the strength of particle–particle connections and polysaccharide–protein interactions when the starch granules expand and absorb water in the continuous phase throughout the heating process. During 14 days of storage, there was no variation in the viability of lactic acid bacteria, except in the sample prepared with 2% tapioca starch, which showed a modest decrease from 6.21 Log10 CFU/g to 5.96 Log10 CFU/g at the end of storage. According to the sensory analysis results, yogurt with 1.0% tapioca starch content increased the overall satisfaction level and the texture rating [134].

Furthermore, the viability of probiotics *Lactobacillus acidophilus* and *Bifidobacterium lactis* in cow milk and soymilk yogurts alternatives were tested after being maintained for 30 days at 10 °C. During storage, *Lactobacillus* increased faster in cow milk yogurts (9.1 Log10 CFU/mL) and soymilk yogurt alternatives (5.4 mL) than *Bifidobacteria* (6.3 and 5.1 Log10 CFU/mL). Consequently, soymilk yogurt alternatives, due to their beany flavor and low viscosity, were less acceptable than cow milk yogurt in taste, texture, and overall acceptability [138]. 

On the other hand, novel ingredients such as peas, lupins, oats, quinoa, and different type of fruits (strawberries, raspberry, blueberries) are also being assessed to improve the physicochemical characteristics of vegan yogurt alternatives and the overall impression of consumers. For example, the physicochemical and sensory characteristics of soy yogurt alternatives with lactic acid cultures (7 Log10 CFU/mL) containing 0.3% gelatin, strawberry flavor, and sweeteners (sucrose, honey, and sucralose) were investigated. The results revealed that the protein level of the honey samples (4.69–4.71%) was higher than control yogurt (3.83%) [135]. Although fructose is the main component in honey, the increased protein content may be explained by the amino acid synthesis that occurs due to fermentation with honey components and yogurt cultures [139]. For the sensorial results, the strawberry-flavored soy yogurt alternative with honey could achieve an acceptable level of quality [135]. Therefore, the inclusion of fruits as a dietary fiber source is insufficient to provide constant physicochemical properties.

### 3.5. Plant-Based Butter Alternatives (PBBAs)

Butter is a semi-solid fat produced mechanically by inverting the dairy cream phase, an oil-in-water emulsion, into a water-in-oil emulsion [140]. Dairy butter with or without salt has 55.2 g/100 g of saturated fat and 222.5 mg/100 g of cholesterol [141]. Despite their technical advantages, saturated and trans fatty acids have been connected to various health issues (cardiovascular disease, hypercholesterolemia, diabetes, and cancer risk) [142,143]. Over the years, alternative plant-derived components and formulations have been proposed and tested to eliminate animal fat, introduce alternate fatty acid arrangements, and include other beneficial compounds in various dietary matrices [140,144]. The consumption of PBBAs (nut and seed butter), an option for dairy butter, has increased considerably [145]. American statistics indicate that peanut butter is the most preferred vegan butter alternative: 299.34 million people consumed it in 2020, and it is expected to increase to 307.7 million in 2024 [146]. 

Recently, new types of PBBAs have been developed to overcome peanut allergies, such as soy, almond, cashew, pistachio, and sesame butter alternatives [147,148,149]. For example, in a comparative study, chemical, microbiological, and sensory evaluations of commercial peanut butter and soybean butter alternatives (cooked, sprouted, and fried) were performed. Peanut butter had the most significant levels of lipids (58.7 g/100 g) and calories (2768 KJ/100 g), whereas soybean butter had the highest levels of moisture (3.7–5.4 g/100 g) and protein (25.8–30 g/100 g). For microbiological testing, sprouted and cooked soybean butter alternatives were more stable than commercial peanut butter. The overall acceptability of fried soybean butter was the lowest (6.3), whereas the sprouted soy butter obtained the best sensory ratings (7.5) [149].

However, to ensure PBBAs’ health and nutritional value, it is essential to pay attention to roasting, gridding, and storage temperatures in the manufacturing process [145]. Therefore, Sanders et al. [150] evaluated consumer acceptability of peanut butter fortified with ground peanut skins (dry-blanched and light- and medium-roasted). Peanut skins addition (2.5 g/100 g peanut butter) yielded peanut butter that matched the control in overall acceptability, independent of heat treatment [150]. Additionally, the paste texture of sesame butter and sesame-milling properties are investigated regarding the possible impact of microwave heating and roll-grinding. Samples roasted in an oven are more inclined to produce aroma than those heated in a microwave oven, and the resulting flavor is characterized by a powerful nut-like odor. Thus, roasting sesame butter in the oven can potentially boost the oil’s volatile content and flavor [151]. 

Moreover, chia seed, sesame seed, watermelon seed, and pumpkin seed in different ratios with the incorporation of olive oil in the formulation of plant-based nutritionally enriched butter were evaluated. The results indicate that the chia + sesame + olive oil formulation butter presents the highest total phenolic content (68.73 ± 0.01 µg GAE/mL), while the chia + watermelon + olive oil butter formulation showed the highest antioxidant activity (52.30 ± 0.01%) [152].

Therefore, PBBAs are good substitutes for dairy butter because nuts and seeds are good sources of protein, fiber, essential fatty acids, and other nutrients [140,141,145,150,151].

### 3.6. Plant-Based Ice Cream Alternatives and Other Types of Sweets

PBIAs (plant-based ice cream alternatives) are becoming more popular due to cow milk’s high fat, cholesterol, and lactose (allergenicity) content. PBIA is recommended for its great nutritional value, particularly in terms of protein content and amino acid balance [153,154,155]. The most challenging problem in the technology of PBIA is the rheological part, especially the viscosity of the cream mixes, which determines the patterns of structure formation [154]. An equally important attribute is the ice cream’s capacity to return to a static condition after freezing [155]. Ice cream mixtures constituted their ideal rheological properties by the use of components able to bind water and structure multi-component mixes (inulin, maltodextrin, gum, polydextrose, and pectin). 

Recently, the impact of enzymatic hydrolysis of protopectin in vegetable purées (table beets, zucchini, broccoli, carrots, and tomatoes) on the structuring ability of ice cream was determined. The benefit of protopectin enzymatic hydrolysis in vegetable purées over acid hydrolysis is that it increases the amount of soluble pectin by 8–12% while using less energy. The enzyme dosage for carrots and beets is highest (0.1–0.2%), and the fermentation lasts from 120 to 180 min to 240 min, whereas with zucchini, broccoli, and tomatoes, the procedure takes 60–120 min, and the enzyme dose is less (0.05–0.10%). Therefore, the efficiency of enzymatic hydrolysis of protopectin varies with the physicochemical properties of vegetables and is greater than that of acid hydrolysis [156].

Additionally, improvements in the rheological, textural, and sensory properties of PBIAs produced with almond and hemp milk by adding dietary fibers (psyllium and pectin fibers 0–10%) were performed. From a technological point of view, to obtain a specific consistency and well-appreciated sensory characteristics, it is recommended that the PBIA contain a maximum of 6% psyllium added and 8% additional pectin fibers. For the organoleptic evaluation, the PBIA with almond milk scored better due to its sweet aroma, while the hemp milk PBIA has been appreciated only for its improved physical-chemical and rheological properties [157].

However, commercially available ice creams often lack beneficial nutrients such as vitamins, natural antioxidants, pigments, and polyphenols. Therefore, Mendonça et al. [158] proposed a functional PBIA using soy extract, soy kefir, and dehydrated jaboticaba peel. Total phenolic component concentrations in the kefir-containing mixtures (6670.40 ± 32.63 mgEAG/100 g) were almost ten times higher than in the soy extract formulation (567.65 ± 35.60 mgEAG/100 g). Moreover, cow, soy, and coconut milk alternatives, as well as combinations of these milk alternatives (25%, 50%, and 75%) were used to compare fermented ice cream with *L. acidophilus* or *B. bifidum*. The probiotic development of *L. acidophilus* (1.29 log10 cfu/g) and *B. bifidum* (1.2 log10 cfu/g) in fermented ice cream was increased (*p* < 0.05) when the mixture of 75% soy and 25% coconut milk alternative was used in replacement of cow milk (0.84 log10 cfu/g) [159].

Along with PBIAs, other vegetable-refreshing desserts are represented by pudding, kulfi, custard, cheesecake, and panna cotta [160,161]. These products are manufactured based on PBMAs and fruits or fruit juices. Due to the growing consumer interest in functional food, Kaur et al. [160] suggested a traditional Indian frozen dessert, kulfi, enhanced with encapsulated betalains extracted from red beetroot pomace. The results indicate a significant improvement in the antioxidant activity (75.27%) and microbial profile (3.14 log CFU/g) compared to the control (28.08%, respectively, 5.11 log CFU/g). Thus, adding fruits and vegetables to PBIA or other vegetable-refreshing sweets during processing might be an effective way of producing functional dairy products with high nutritional content [157,158,160,161].

**Table 3 foods-12-01883-t003:** Nutritional values/100 g *.

Dairy Product/Alternative	Source	Water (g)	Protein (g)	Total Lipid (g)	Carbohydrate (g)	Calcium (mg)	Energy (kcal)
Milk/alternative							
Whole (3.25% fat)	Bovine	88.1	3.27	3.2	4.63	123	61
Unsweetened, plain, refrigerated	Almond	96.5	0.66	1.56	0.67	158	19
Unsweetened, plain, refrigerated	Oat	90.6	0.8	2.75	5.1	148	48
Unsweetened, plain, refrigerated	Soybean	91.5	2.78	1.96	3	155	41
Cheese/alternative							
Ricotta, whole milk	Bovine	72.9	7.81	11	6.86	224	158
Feta, whole milk	Bovine	51.9	19.7	19.1	5.58	371	273
Curd cheese	Soybean	70.9	12.5	8.1	6.9	188	151
Tofu fried	Soybean	50.5	18.8	20.2	8.86	372	270
Tofu salted and fermented (fuyu)	Soybean	70	8.92	8	4.38	46	116
Yogurt/alternative							
Plain, whole milk	Bovine	85.3	3.82	4.48	5.57	127	78
Tofu	Soybean	77.5	3.5	1.8	16	118	94
Butter/alternative							
Stick, unsalted	Bovine	17.4	-	81.5	-	14	-
Creamy	Almond	1.75	20.8	53	21.2	264	645
Crunch style, without salt	Peanut	1.14	24.1	49.9	21.6	45	589
Without salt	Cashew	2.34	12.1	53	30.3	61	609

* Source: USDA Food Composition Database [162].

## 4. Health Benefits of PBDAs and Research Gaps

There is a variety of clinical evidence that supports the positive effects of a plant-based diet, especially when it comes to the prevention but also the amelioration of specific chronic health conditions such as type 2 diabetes, dyslipidemia, metabolic syndrome, obesity, cardiovascular diseases, hypertension, and even several types of cancer [163,164,165]. This fact does not presume or encourage the total exclusion of animal-based food products from diets, but a well-balanced plant-animal-based diet for an optimal health state following the metabolic needs of each individual [163,166,167,168,169]. Over and above that, more and more interest is put in encouraging people in adopting a plant-based diet as part of their lifestyle. Moreover, they propose a sustainable and economic point of view, because of the elevated energy resource consumption for the production of animal-based food products compared with those based on vegetable sources [170].

### 4.1. Metabolic Diseases

Metabolic diseases affect an increasing number of individuals, and many affected people are associated in multiple cases with an imbalanced diet and lifestyle [168]. For instance, the prevalence of metabolic syndrome including obesity, type 2 diabetes, and cardiovascular illnesses are tightly linked with the impaired glucose metabolism that is induced by a predominant animal-based products consumption, while a shift to a preponderant consumption of vegetable-based products may lead to positive effects on glycemic control in affected individuals [171]. The metabolic cardiovascular-associated risks including hypertension, dyslipidemia, hyperuricemia, hypercholesterolemia, and obesity were shown to express a low-to-moderate prevalence in participants that were evaluated with a dominant plant-based diet [171,172,173].

So, in line with the multiple advantages of vegetable food product consumption over the general human health state, the PBDAs are more and more in the sight of adult people for replacing conventional animal-based dairy products [3]. In a study conducted by Schiano et al. (2020) [3] regarding consumers’ perceptions of the use of PBDAs for their health-related properties, the authors showed a growth in their perception, acceptability, tolerance, and daily consumption. The commercially available PBMAs obtained from different cereals or oil seeds (examples from Section 2 and Section 3) provide a wide range of micro- and macro-components with direct metabolic implications [174]. For instance, alternative dairy beverages obtained from soybean processing were demonstrated to exert beneficial effects in overweight and obesity control, along with positive effects in alleviating symptoms associated with premenopausal syndrome and osteoporosis in postmenopausal women [175,176]. Moreover, the beneficial properties of soybean-based beverages were proved by a clinical trial (participants aged 35–68 years) where the effects of soy milk alternatives supplemented with probiotic strains (*L. plantarum* A7) in type 2 diabetic patients were investigated [177]. The study’s outcome revealed that the patients who consumed 200 mL/day of probiotic soy milk alternative for 8 weeks exhibited a decrease in the promoter methylation in proximal and distal MLH1 promoter region compared with the baseline values. In addition, a significant increase in superoxide dismutase activity was noticed in the probiotic soy milk alternative group compared with the baseline value. The same study points out that the consumption of soy milk alternatives together with probiotics strengthens the antioxidant status in type 2 diabetic patients, supporting once again its importance as a promising agent for diabetes management [177].

Plant-based dairy drinks such as oat milk alternatives have health benefits in alleviating diabetes-associated symptoms and hyperlipidemia [178]. To support the statement, an experimental study conducted on 42 diabetic rats evidenced the therapeutic potential of both fermented and unfermented oat milk alternatives. The study’s main outcome showed an improvement in intestinal microbiota diversity and pointed out the lower risk of developing hyperglycemia and hyperlipidemia associated with type 2 diabetes mellitus [178]. In addition, PBDAs based on oat lower the serum and LDL cholesterol levels after moderate consumption, facts evidenced in a study conducted on 52 free-living men with moderate hypercholesterolemia who consumed 750 mL/day of oat milk for 5 weeks [179]. As oat milk alternatives contain high amounts of dietary fiber such as β-glucan, they can serve as an efficient cholesterol-reducing tool and also assist in the management of the metabolic syndrome of adult patients [179,180,181].

PBDA drinks such as hemp milk alternatives could be used in the management of cholesterol and triglycerides in adult individuals affected by obesity [182]. As hemp is highly rich in easily digestible complete proteins, PUFAs (e.g., linoleic, linolenic, stearidonic, and gamma-linolenic acids), and essential amino acids (e.g., arginine, aspartic acid, and glutamic acid), alternative dairy products based on hemp bring valuable nutrients to the consumers and could help in the prevention and treatment of metabolic-associated affections such as obesity. For instance, in a study conducted on Winstar female rats was observed that the consumption of hemp milk alternative instead of water significantly reduces the level of serum triglycerides and total cholesterol [183]. On the other hand, a few downsides must be considered while consuming hemp-derived products, as some of the bioactive from hemp milk interfere with the thyroid hormones by decreasing them, a fact observed in animal studies [103,183].

PBDAs highly rich in lipids such as coconut milk or groundnut milk alternatives also have positive outcomes considering metabolic diseases in adult humans. Coconut milk alternative, for instance, was proved to be effective in reducing LDL and raising HDL cholesterol levels, while groundnut-based milk alternative was associated with a reduction of serum triglycerides and LDL cholesterol in healthy adults with overweight or incipient obesity [184,185].

Nonetheless, PBDAs directly influence the metabolic profile of humans, bringing valuable bioactives that could intervene in the prevention and treatment of certain metabolic-associated disorders. Still, a plant-based diet is encouraged to be part of the lifestyle, but without neglecting the consumption of animal products to achieve a balanced diet that covers most of the essential nutritional components.

### 4.2. Dermatological Diseases

There was a long-held belief that nutrition had no bearing on several prevalent dermatological diseases, and the importance of nutrition has historically been a neglected part of the treatment [186]. However, recent research has revealed a significant association between diet and skin health, and in some cases, dietary adjustments can influence how a skin disease progresses (e.g., acne, skin aging, psoriasis) [186,187]. Moreover, systematic medications prescribed for dermatological diseases are known to increase the risk of other associated diseases (e.g., patients with psoriasis present a higher risk for cardiovascular disease) [187].

Acne has a complex pathophysiology which might be significantly influenced by factors such as genetics, environment, and hormones [188]. Excessive sebum production, *Cutibacterium acnes* hyperproliferation, hyperkeratinization of the pilosebaceous follicles, and inflammation are also important elements in the pathogenesis of acne. The overproduction of sebum may be caused by increased androgenic hormones and insulin-like growth factor 1 (IGF-1) activity [189,190]. Moreover, according to recent studies, nutrition represents a potential contributor to the progression of acne due to its involvement in various pathogenetic mechanisms [189].

More precisely, several recent studies have offered compelling evidence of a link between dairy product consumption and acne [189,191,192]. For acne patients, avoiding dairy products may be beneficial, particularly regarding the casein content in cow’s milk, which increases the IGF-1 [191]. Additionally, consuming whole and skimmed cow’s milk may worsen acne since it includes more hormones or other bioactive molecules (α-lactalbumin, growth factor-stimulating hormones, steroids, IGF-1, etc.) [189]. Even though it is difficult to evaluate the wide variety of processed dairy products, such as cheese, some foods containing cow’s milk, such as ice cream, have also been associated with acne [189]. Thus, skim milk was proved to be more associated with acne than other dairy products. A practical alternative for acne patients may be the consumption of PBDAs. For instance, soy-based dairy products have been shown to decrease the incidence of acne. Studies suggest that isoflavones and phytoestrogens contained in soy and other PBDAs inhibit androgen-induced sebum production, which has been shown to improve the reduction of acne lesions [188,189,190].

On the other hand, the skin is the human body’s largest organ and serves as a barrier to protect internal organs and cells from external factors [193]. Skin health and aging are both influenced by intrinsic (e.g., skin thickness, microvasculature structure, sex hormones) and extrinsic (diet, sleep, humidity, UV radiations) mediators [193,194]. Additionally, unwanted dermal changes occur with aging due to decreased estrogen levels. However, the ingestion of estrogen or phytoestrogens facilitates the reconversion of the skin. These phytoestrogens (e.g., genistein, daidzein, and glycitein) are included in soy milk alternatives and other plant-based products. They have a similar structure to estrogen, being included in the category of isoflavones. Genistein bonded to the ß-estrogen receptor restores dermal breakdown. Likewise, daidzein has strong antioxidant, anti-aging, and anti-inflammatory properties. Despite its lower concentration in soy milk alternatives and lower binding activity to estrogen receptors, glycitein has a greater estrogenic response considering its higher bioavailability [189,193]. Additionally, it was shown that the consumption of unsweetened almond milk alternatives stimulates the production of collagen, improving skin health [81]. Tiger nut milk alternative contains vitamin E, which helps to prevent cell aging, increase skin elasticity, and reduce the appearance of wrinkles [195].

Psoriasis is an autoimmune condition that causes skin cells to build up and form lesions on the skin. Food is one of many potential triggers that may worsen a person’s psoriasis symptoms or flare [196]. Therefore, the category of dairy products is also included among the foods to be avoided by psoriasis patients due to the high content of arachidonic acid, which irritates the intestinal mucosa, prolonging psoriasis outbreaks [197]. Based on these facts, more studies are required to confirm a link. Moreover, research suggests that dairy may cause inflammation, and avoiding dairy products was found to be beneficial for controlling psoriasis [198]. However, in the case of psoriasis disease, the consumption of almond milk alternatives may help to ease and calm the symptoms due to its composition of vitamins, antioxidants, and anti-inflammatory agents [81].

The consumption of PBDAs may be beneficial in various dermatological conditions (acne, skin aging, psoriasis, etc.). Due to the high composition of bioactive compounds and their involvement in pathogenetic mechanisms, PBDAs represent a great alternative to cow milk for patients with different dermatological diseases.

### 4.3. Degenerative Arthritis, Osteoarthritis, and Rheumatoid Arthritis

Rheumatic disease is an umbrella term that refers to arthritis and several other conditions such as lupus, gout, scleroderma, and spondyloarthropathies that affect the joints, tendons, ligaments, bones, and muscles. Osteoarthritis (OA) is sometimes referred to as degenerative arthritis or degenerative joint disease, and it occurs most frequently in the hands, hips, and knees. With OA, the cartilage within a joint begins to break down and the underlying bone begins to change. Rheumatoid arthritis (RA) is a chronic, immune-mediated inflammatory disease with articular manifestations that is often accompanied by systemic comorbidities, and it has a complex etiology. The main difference between OA and RA is the cause behind the joint symptoms. While OA is a common age-related chronic condition that is most frequently a consequence of overweight/obesity that affects the joints through mechanical loading, RA is an autoimmune disorder characterized by symmetric, erosive synovitis and, in some cases, extraarticular involvement [199].

Shoenfeld and Isenberg, in 1989, determined the “mosaic of autoimmunity” as the interplay between genetic, hormonal, immunological, and environmental factors in the pathogenesis of autoimmune diseases, including RA [200], and several scientific reports indicate a potential link between dietary factors and alterations in epigenetic pathways [201]. Thus, the possible effects of environmental factors on fundamental biological processes and the etiology of autoimmune diseases are hypothesized. In this regard, a recent literature review on RA highlights the proposal of the European League Against Rheumatism to incorporate lifestyle interventions in the multidimensional approach of the current RA management through a combination of physical exercise, optimum nutrition, social support, and self-management strategies that may help in controlling potential inflammatory triggers and improve the overall quality of life [202].

According to a study conducted on the design of an anti-inflammatory diet for patients with RA, there is evidence that some ingredients have pro- or anti-inflammatory effects, and recent scientific literature shows that both diet and the gut microbiome are linked to circulating metabolites that may modulate inflammation [203]. It is also important to mention that the gut microbiota is an environmental factor that influences metabolic and immune homeostasis and RA patients exhibited gut microbial dysbiosis before the onset or at diagnosis of the disease [204]. Furthermore, a gut microbial imbalance, characterized by the loss of metabolically and immunologically beneficial bacteria and a concomitant increase in potentially pathogenic microbes (pathobionts), is associated with several chronic inflammatory syndromes [205]. For this reason, nutritional interventions based on increased consumption of fiber, as a preponderant part of vegetarian and vegan diets, can improve the gut microbiome, increase bacterial diversity, and subsequently reduce inflammation and arthralgia [206].

A comprehensive review of nutrition and its role in the onset of RA highlights that the consumption of long-chain omega-3 PUFA is associated with a reduced risk of RA, probably due to their anti-inflammatory properties [207]. Therefore, PBDAs belonging to seed-based sources such as sesame, flax, hemp, and sunflower that are rich in omega- 3 fatty acids could represent a reliable source of intake. Along the same line, beverages are of fundamental importance in dietary habits, often playing a role that goes beyond their simple role of rehydration. For example, studies on the health benefits of plant-based food components in knee or hip OA included soy milk alternatives once a day vs. less than once a day in the cross-sectional studies and the models suggested a significant inverse association between soy milk alternative intake and osteophytes [208].

In conclusion, RA is an autoimmune disease and OA is an age-related chronic condition, and both have common inflammatory reactions that can be partially modulated by nutritional patterns that integrate plant-based foods, including PBDAs, to control weight gain and reduce systemic inflammation.

## 5. Special Dietary Needs for Plant-Based Dairy Alternative Consumers

### 5.1. Pregnancy and Infancy

Dairy products from cow milk are avoided for several reasons, the most important being medical reasons such as lactose intolerance, cow’s milk protein allergy (CMPA), cholesterol issues, or concerns regarding growth hormones or antibiotic residues in the milk. Nonetheless, plant-based proteins are generally considered to have a lower nutritional quality in comparison to animal-derived proteins, due to their composition of amino acids and their absorption and utilization by the body. Additionally, the processing method has an important influence on the final composition of the product [209]. The recommended protein content for plant-based beverages that are fortified is at least 24 g of proteins per liter of drink [210].

Besides proteins, plant-based beverages present other nutritional differences compared to cow milk such as lipidic profile, lower energy, different glycemic index, and lower levels of vitamins (B12, B2, D, and E) and minerals, especially calcium. Another important factor that differs from cow milk is the bioavailability of the different nutrients, even those added for the fortification of the drinks. This plethora of differences makes plant-based beverages an unsuitable substitute for cow milk and cow milk-based formulas, especially for infants aged below 24 months, only in particular medical cases and with proper fortification. The health benefits of plant-based beverages are not yet fully known, however, the different nutritional composition of the drinks poses concerns, as inadequate substitution of cow milk or formulas with plant-based drinks leads to nutritional deficits.

CMPA is an immune system reaction to one or more milk proteins that leads to an inflammatory response while presenting a 2–6% prevalence in infants and 0.1–0.5% in adults. Complete removal of cow milk from the diet is the only treatment for this disease [209], but it brings nutritional problems such as a lack of protein, fat, calcium, phosphorus, and vitamin B12 for infants. The best-known alternative in this situation is maternal breastfeeding, as it is considered the optimal source of nutrients for infants [211].

However, if breastfeeding is not possible for infants suffering from CMPA, PBDAs could only pose a viable substitution alternative for infants above 2 years, as the drinks lack crucial nutrients needed for younger infants. It is important to raise awareness regarding the nutritional properties and composition of different plant-based beverages for a better choice for children with CMPA [212].

### 5.2. Elderly

The global population is progressively aging. Efforts to support this age group to sustain vitality could have far-reaching ramifications for both health and healthcare costs [213]. The number of older adults, ≥60, will increase by 34% by 2030, and in 2019 there were 1 billion people in this age group [214]. Frailty is an elderly illness that increases sensitivity to stress and boosts the risk of undesirable health consequences such as incapacity, hospitalization, or death [215]. In consequence, during the next years, health organizations will confront an increasing burden of age-related illnesses [216]. However, on the other hand, these can be prevented and possibly reversed through actions such as dietary improvements as well as increasing physical activity [217,218].

In the last decades, dietary approaches to preventing frailty are becoming more popular, and until now, the majority of the scientific literature considered that increased vegetable and fruit intake along with increased adherence to diets high in fruits, vegetables, legumes, and grains and low in red and processed meat have all been attributed to a reduction in the incidence of frailty [219,220,221].

PBDAs are now being considered as possible substitute products for animal-based dairy products due to changing market demands to maintain a balanced diet. Therefore, more research should focus on improving the safety and quality of these alternative diet sources. Moreover, to achieve a PBDA with certain organoleptic properties that are safe for the elderly population consumption, technical-functional aspects and manufacturing factors are crucial [222]. In general, the majority of the PBDAs’ functional properties are associated with strong support during frailty, the quality of those is crucial since improper plant diets could have a higher negative influence on aging [214].

Along with advancing age, the phenomenon of muscle loss and strength with severe consequences in the elderly is extremely well known [223]. Resistance exercise combined with protein consumption provides an anabolic stimulus for skeletal muscle protein synthesis; however, a higher protein intake may affect elderly adults with undiagnosed chronic kidney disease; therefore, a plant-based dairy diet must be carefully structured and adapted to their specific needs [224].

Even though PBDAs are becoming increasingly popular, even among the elderly, there are still concerns regarding their nutritional aspects, especially calcium, vitamin D, and B12 supplementation concentrations, as specific nutritional standards must be met for these types of products [225]. Likewise, with this trend of PBDAs, it is possible that the older population will not increase their dose of plant sources rich in protein and other essential nutrients as recommended and only increase the present intake of less healthful plant-based diets [168]. More research is needed to properly understand the impact of increasing particular types of PBDAs on diet quality and overall wellness, as well as the impact of these dietary changes on the aging population [226].

### 5.3. Allergies, Lactose Intolerance

The most common adverse reactions associated with milk consumption are CMPAs and LI. Allergies and intolerances result from the body’s inability to digest, absorb, and metabolize a specific component [227]. LI is the body’s inability to digest lactose due to the total or partial absence of lactase, which is an enzyme specialized in this action [228]. Alongside LI is CMPA, which is characterized by immune reactions when the body encounters cow milk protein [229]. Thus, the demand for PBMAs has increased over the years [230].

LI, also known by the term lactose malabsorption [231], is an irreversible clinical syndrome characterized by specific signs and symptoms following the consumption of lactose, a disaccharide [232]. Lactose can be found in dairy products, milk, and mammalian milk [228]. Once lactose is ingested, it is normally hydrolyzed into glucose and galactose by the lactase enzyme found in the small intestine brush border [233]. Due to primary and secondary causes, lactase deficiency induces clinical symptoms such as abdominal pain, bloating, flatulence, and diarrhea [234,235]. However, some extraintestinal symptoms related to LI include headache, vertigo, and memory impairment [232]. The severity of LI varies from one individual to another. The medical nutrition therapy recommended for patients suffering from LI is a low-lactose diet.

On the other hand, CMPA is an immune-mediated reaction that appears during the first year of life and tends to remit in childhood [232,235]. The gastrointestinal and extraintestinal symptoms that appear in CMPA can be mediated by immunoglobulin E (IgE) or non-IgE or by mixed reactions. The symptoms that are IgE-mediated usually occur affecting the skin (hives, angioedema, etc.), and the respiratory system (nasal itching, sneezing, cough, etc.); they may also be associated with diarrhea, abdominal pain, vomiting, or even signs and symptoms of anaphylaxis [228,232]. The non-IgE-mediated symptoms may be represented by vomiting, diarrhea, blood and/or mucus in the stools, abdominal pain, malabsorption of nutrients (poor weight gain), and atopic eczema. The dietary treatment suggested for patients with CMPA is a diet free of cow milk proteins [232].

Issues such as LI and CMPA have compelled some specific populations to look for milk alternatives that are more or least nutritionally equivalent to conventional milk [81,230]. Besides that, PBDAs satisfy the dietary needs of these patients, offer a wide and diversified assortment of products, and are inexpensive alternatives. Currently, the market for milk substitutes is represented by soybean, oat, coconut, cocoa, multigrain milk alternatives, and other derived products [230]. Additionally, most of these alternatives are produced through controlled fermentation due to their functional bioactive composition. These PBDAs are appreciated for their functionally active compounds, which are often linked to disease-prevention characteristics and health-promoting properties [233]. One significant advantage of analogs over conventional milk is that the energy input per unit of milk alternative produced is much lower compared to animal milk. Moreover, PBDAs always offer the opportunity for fortification and their composition can be enhanced based on the patient’s needs (addition of vitamins, minerals, etc.) [230].

## 6. Conclusions

Broadly translated, our findings indicate that plant-based dairy alternatives have a significant role in human health, both for consumers who choose this type of product out of their conviction and consumers conditioned by different factors, such as allergies. Therefore, this review paper focused on the interrelational line starting from unprocessed sources, such as cereals and legumes, continuing with the obtained products and their effect on the body.

Regarding the raw materials and the products obtained from these complex matrices (e.g., soybeans and soy milk alternatives), both need a comprehensive understanding, starting from the nutritional profile and up to the physicochemical properties of the final products and the technological processes that must be applied and optimized, especially their shelf life and consumer acceptance.

Plant-based dairy alternatives have a wide range of involvement, from vegetarians and vegans to various illnesses or stages of life. In this review, the positive effects that can be achieved on conditions such as rheumatoid arthritis, metabolic and dermatological diseases, and other chronic illnesses were highlighted. However, it was also necessary to draw attention to some other critical topics, such as the consumption of these alternatives for newborns.

Future investigations are necessary to validate the kind of conclusions that can be drawn from this study, to clarify the aspects regarding the processing, similarity in the nutritional profile to achieve the same nutrient targets as animal-based dairy products, and consumer acceptance, which is a significant pillar from the economical point of view.

Regarding groups with special dietary needs, starting from pregnant women and the elderly, a wider analysis involving multidisciplinary factors, such as medical, nutritional, psychological, and social ones, is necessary. The group that includes pregnant women, newborns, and child growth is a more sensitive group that requires increased attention, both from a medical and legislative point of view.

In summary, this paper argued that plant-based dairy alternatives are in continuous development, and the sources from which they can be obtained are numerous, offering valuable nutrients to human health. The correlations between different pathologies and the consumption of plant-based dairy alternatives are beneficial, improving various parameters, but more studies are needed to clarify the long-term effects and the administration for those special dietary needs.

## Figures and Tables

**Figure 1 foods-12-01883-f001:**
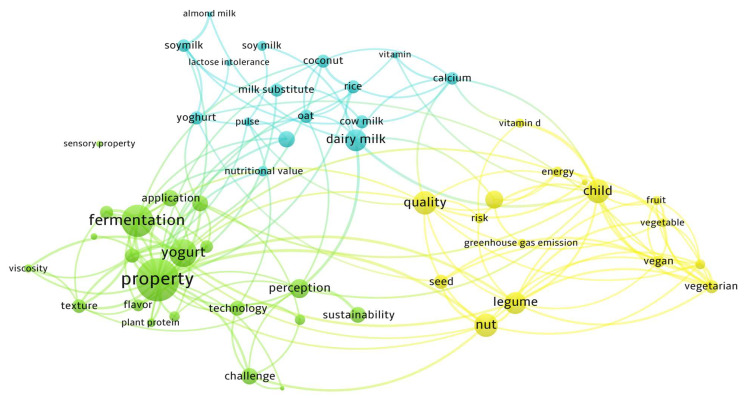
Plant-based dairy alternatives—related keywords, such as cow milk, fermentation, sustainability, legumes, vegetarian, nutritional value, and perception (VOSviewer version 1.6.18).

**Figure 2 foods-12-01883-f002:**
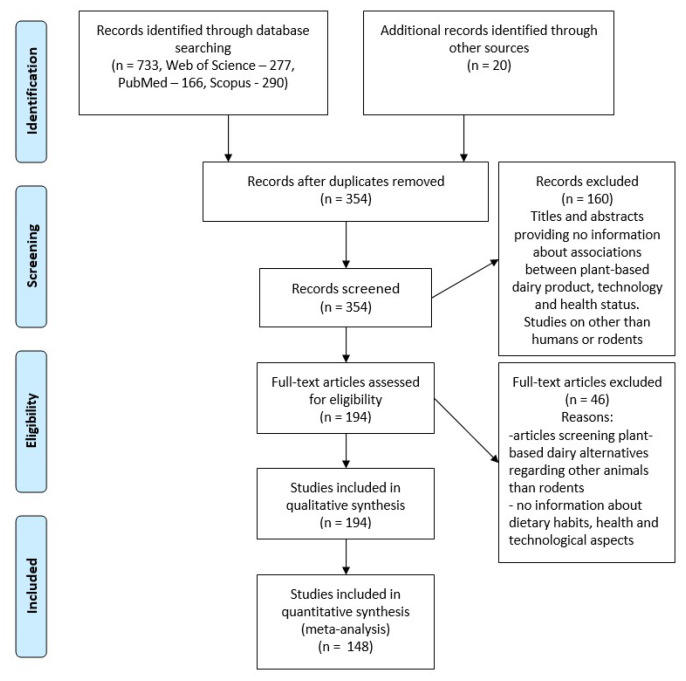
Preferred Reporting Items for Systematic Reviews and Meta-Analyses (PRISMA) flow diagram of the study selection process.

## Data Availability

Data are contained within the article.

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
