# Peer review of "Plant-Based Dairy Alternatives—A Future Direction to the Milky Way"

_foods, 2023, doi:10.3390/foods12091883_

Round 1

Reviewer 1 Report

Comments and recommendations:

  • I do not agree with the title, especially with the word “new” (I would use “possible” or “future” with “?”. Please, try to change the title.
  • The manuscript is not easy to read. Long sentences. Unnecessary information (for example, 132-133, 140). 
  • Moreover, the authors should use more paragraphs regularly. For example, the chapter 2.1. This chapter has almost two pages with two paragraphs! It would be appropriate (also for better reading) to use a new paragraph for every plant source (for example, a new paragraph for l. 140 Rice, l.149 Sweet corn, l. 57 Quinoa). The same applies to other chapters (2.2, 3.1. …). 
  • Table 1 (nutritional profile of plant sources) is very interesting. In addition, the authors should also create a new table with information on positive vs negative compounds and properties that characterize every (or selected) plant source. Thus, the text of the manuscript becomes more precise (simpler and clearer).
  • I also see a problem in the use of abbreviations (too much and confused) 

- please do not use abbreviations for cow milk (CM); 

- please, clarify/unify abbreviations for plant-based products (for example, PBMAs and PBDAs - they are the same); 

- sometimes you use abbreviations PBDAs (l. 33), sometimes the whole term "plant-based dairy products" (for example, l. 157, 162)

- please do not use abbreviations in Conclusions   

Author Response

Dear Reviewer #1, we kindly thank you for revising our manuscript and for your feedback. We have considered your suggestions and some changes were implemented into the manuscript, as explained below:

Comment 1: I do not agree with the title, especially with the word “new” (I would use “possible” or “future” with “?”. Please, try to change the title.

Answer: Thank you for this valuable comment. We modified the title as follows:” Plant-based dairy alternatives – a future direction to the milky way”. All the changes can be seen with track changes.

Comment 2: The manuscript is not easy to read. Long sentences. Unnecessary information (for example, 132-133, 140).

Answer: Thank you very much for your observation. We have adapted to your comment by deleting unnecessary information.

Comment 3: Moreover, the authors should use more paragraphs regularly. For example, the chapter 2.1. This chapter has almost two pages with two paragraphs! It would be appropriate (also for better reading) to use a new paragraph for every plant source (for example, a new paragraph for l. 140 Rice, l.149 Sweet corn, l. 57 Quinoa). The same applies to other chapters (2.2, 3.1. …).

Answer: We kindly thank you for this remark. Chapters 2.1, 2.2, 3.1, and the other chapters throughout the manuscript have been modified according to your remark, by dividing them into more paragraphs.

Comment 3: Table 1 (nutritional profile of plant sources) is very interesting. In addition, the authors should also create a new table with information on positive vs negative compounds and properties that characterize every (or selected) plant source. Thus, the text of the manuscript becomes more precise (simpler and clearer).

Answer: Thank you for this useful recommendation. A new table has been introduced in the manuscript, named ‘Table 2. Beneficial and anti-nutritional effects of PBDAs compounds.

Comment 4: I also see a problem in the use of abbreviations (too much and confused) 

- please do not use abbreviations for cow milk (CM); 

- please, clarify/unify abbreviations for plant-based products (for example, PBMAs and PBDAs - they are the same); 

- sometimes you use abbreviations PBDAs (l. 33), sometimes the whole term "plant-based dairy products" (for example, l. 157, 162)

- please do not use abbreviations in Conclusions   

Answer: We kindly thank you for all these valuable observations. The abbreviation for cow milk has been modified throughout the manuscript. The abbreviation for plant-based dairy alternatives and plant-based dairy products have been unified, and abbreviations in the conclusions have been excluded.

Reviewer 2 Report

This review tries to combine the food processing and food uses for plant-based dairy alternatives with the medical issues that might be related to the use of these products.  In the end, it accomplishes neither. I recommend eliminating the vague discussion of the medical issues for using these products (the review is far too superficial in these areas) and focusing solely on the foods products and their use, with a comprehensive evaluation of which plant products are being utilized and why these products are appropriate.  Move this paper from the superficial to a more indepth focus.  It could be a great contribution to the literature.  

Author Response

Comment: This review tries to combine the food processing and food uses for plant-based dairy alternatives with the medical issues that might be related to the use of these products.  In the end, it accomplishes neither. I recommend eliminating the vague discussion of the medical issues for using these products (the review is far too superficial in these areas) and focusing solely on the foods products and their use, with a comprehensive evaluation of which plant products are being utilized and why these products are appropriate.  Move this paper from the superficial to a more indepth focus.  It could be a great contribution to the literature.  

Dear Reviewer #2,

We kindly thank you for your valuable recommendations. We reexamined the chapter regarding medical issues, and then we tried to switch the focus to a more in-depth perspective on the first part of the manuscript.

Reviewer 3 Report

This article Plant-based dairy alternatives - a new direction to the milky 2 way) is good, however here are a few suggestions that will improve the quality of the manuscript if followed by the authors

1.                  Add some numerical values in the abstract section.

2.                  Add proper conclusive line in the end of the abstract.

3.                  It seems nice if the abstract started with some introductory lines about plant-based dairy (PBD), Overall, the abstract is fine but it needs to be a bit more focused on the aim and methodology sentences, and add conclusion part in the abstract.

4.                  Keywords should be written in alphabetical order.

5.                  Keep the introduction with recent supportive findings, if you can find some relevant intro of the main title recently published in 2022, that could be much better.

6.                  Add importance and reasoning of this study in the form of rationale in the end of introduction section.

7.                  The methodology used for the bibliographic search and the criteria for the selection of the articles should appear at the end of the introduction

8.                  Verify that all scientific names are in italic throughout the whole document.

9.                  The use of the English language (syntax, grammar, etc.) and details need to be revised in order for the manuscript to be accepted.

10.              The descriptive text needs to expand on the concepts and findings of the papers cited: The authors should enhance this element. In addition, each paragraph only contains fewer citations. They appear insufficient for a review that tries to Plant-based dairy alternatives benefits. Please incorporate further research and describe it in the manuscript. The descriptive text needs to expand on the concepts and findings of the papers cited: The authors should enhance this element. In addition, each paragraph only contains one to four citations. Please incorporate further research and describe it in the manuscript.

11.              Grammatically, it needs a lot of serious attention by authors or any native member.

Author Response

Dear Reviewer #3,

we kindly thank you for taking the time to thorough revision our manuscript. Your recommendations were very valuable to us, and they improved the quality of our work. We have considered your suggestions and the manuscript was amended accordingly. In the revised version of the manuscript, you can observe the modifications, which we strongly believe would satisfy the requirements. Also, if is anything else to be addressed, we would very happy to apply if there is something else.

Comment 1: Add some numerical values in the abstract section.

Answer: Thank you very much for this suggestion. Numerical values have been introduced as follow (line 17 – 18): “To achieve the desires of individuals who consume plant-based dairy alternatives (PBDAs), the production tendency of it is increasing, with a predictable rate of over 18.5 % in 2023 from 7.4 % at the moment.”

Comment 2: Add proper conclusive line in the end of the abstract.

Answer:  Thank you for this useful recommendation.  A proper conclusive line has been introduced as follow (line 28 – 31): “This paper tries to provide an overview of plant-based milk alternatives from the technological point of view used in the current moment and future perspectives to improve the quality of the final products, as well as the research that requires more studies to clarify the beneficial and less beneficial effects in human health.”

Comment 3: It seems nice if the abstract started with some introductory lines about plant-based dairy (PBD), Overall, the abstract is fine but it needs to be a bit more focused on the aim and methodology sentences, and add conclusion part in the abstract.

Answer: Thank you for your recommendation. New lines have been introduced in the abstract regarding the aim, methodology, and conclusion.

    Comment 4: Keywords should be written in alphabetical order.

    Answer: Thank you for this valuable comment. The keywords have been written in alphabetical order as follows: dairy, fermentation, food processing, legume, milk, nutritional value, nuts, sustainability, vegetarian.

Comment 5: Keep the introduction with recent supportive findings, if you can find some relevant intro of the main title recently published in 2022, that could be much better.

Answer: Thank you for your suggestion. New references from 2022 have been introduced in the introduction.

Comment 6: Add importance and reasoning of this study in the form of rationale in the end of introduction section

Answer: Thank you for this remark. A new paragraph has been introduced at the end of the introduction section (lines 127 – 134).

Comment 7: The methodology used for the bibliographic search and the criteria for the selection of the articles should appear at the end of the introduction

Answer: Thank you very much for this valuable suggestion. A methodology for the selection of the articles has been introduced as seen in the manuscript: lines 126 – 150.

Comment 8: Verify that all scientific names are in italic throughout the whole document.

Answer: Thank you for this valuable observation. The paper was modified for all the scientific names in italic.

Comment 9: The use of the English language (syntax, grammar, etc.) and details need to be revised in order for the manuscript to be accepted.

Answer: Thank you for this valuable suggestion. The English language has been revised.

Comment 10: The descriptive text needs to expand on the concepts and findings of the papers cited: The authors should enhance this element. In addition, each paragraph only contains fewer citations. They appear insufficient for a review that tries to Plant-based dairy alternatives benefits. Please incorporate further research and describe it in the manuscript. The descriptive text needs to expand on the concepts and findings of the papers cited: The authors should enhance this element. In addition, each paragraph only contains one to four citations. Please incorporate further research and describe it in the manuscript.

Answer: Thank you very much for your comment. The manuscript has been improved accordingly to your suggestions, new paragraph, table, and references have been introduced throughout the manuscript.

Comment 11: Grammatically, it needs a lot of serious attention by authors or any native member.

Answer: We kindly thank you for this comment. A native member has improved the grammatical issues.

Round 2

Reviewer 1 Report

What difference is between plant-based dairy alternatives (PBDAs) and plant-based milk alternatives (PBMAs)??? Please use only one abbreviation, be consistent (l. 38-41)

Author Response

We would like to take this opportunity to express our sincere thanks for identifying areas of our manuscript that needed corrections or modifications. The suggestions and recommendations helped us to improve the quality and content of the paper. We appreciate the time dedicated to reading and reviewing our manuscript. The difference between plant-based dairy alternatives (PBDAs) and plant-based milk alternatives (PBMAs) is that PBDAs include all the dairies mentioned in the paper (milk, cheese, yogurt, cream, and desserts), and PBMAs refer only to the milk alternatives, not including the others. Most of the PBDAs are produced from milk, dairy being a much broader group of food products. As we wanted to separate the terms, to split them into different categories, we used these two different terms, PBDAs, and PBMAs.

Reviewer 2 Report

The paper covers too much ground in too superficial a manner.  A focus on the technical aspects of producing plant based beverages would be appropriate.  A focus on the market aspects of these beverages would also be appropriate.  The health benefit data is just not available.  You are trying to cover too much ground superficially in a VERY long paper that few are likely to read.  

Author Response

Dear Reviewer #2,

We would like to take this opportunity to express our sincere thanks for identifying areas of our manuscript that needed corrections or modifications. The suggestions and recommendations helped us to improve the quality and content of the paper. We appreciate the time dedicated to reading and reviewing our manuscript.

As you mentioned, the manuscript already covers too much ground, the technical aspects are mentioned in each subchapter per each type of dairy product, and the market aspects of these beverages were not the focus of our paper. The chapter regarding health benefits has been restructured and shortened, excluding unnecessary information.

Reviewer 3 Report

Authors have improved the article in well manners 

I just have one concerned about its role in dairy industry.

Is there any toxicity due to plants or any fertilizers?

Author Response

Dear Reviewer #3,

We would like to take this opportunity to express our sincere thanks for identifying areas of our manuscript that needed corrections or modifications. The suggestions and recommendations helped us to improve the quality and the content of the paper. We appreciate the time dedicated to read and review our manuscript.

Research has shown that some of these alternatives, such as soy milk and almond milk, can contain trace levels of heavy metals such as cadmium and lead, which can be potentially harmful to human health if consumed in significant amounts over a long period of time. The levels of these elements are generally within safe limits, but it is important to maintain a varied and balanced diet. Unfortunately, as the paper covers already many aspects, this concern is proposed for further study.
